# Assessment of Antioxidant Contents and Free Radical-Scavenging Capacity of *Chlorella vulgaris* Cultivated in Low Cost Media

**Kulwa Mtaki \*, Margareth S. Kyewalyanga and Matern S. P. Mtolera**

Institute of Marine Sciences, University of Dar es Salaam, P.O. Box 668, Zanzibar 71116, Tanzania; maggie@ims.udsm.ac.tz (M.S.K.); mtolera@ims.udsm.ac.tz (M.S.P.M.)

\* Correspondence: mtakikulwa@yahoo.com; Tel.: +255-759-226268

**Abstract:** The current study assessed antioxidants contents (total phenolics and total flavanoids, β-carotene and lycopene) present in *Chlorella vulgaris* (*C. vulgaris*) cultivated in low cost media and their free radical scavenging activities. Microalgae was cultured using Bold basal medium (BBM) as a control, 5% banana stem compost (BCM) and aquaculture wastewater supplemented with 1.0 g/L NPK (ANM). The free radical scavenging ability was analysed using 1,1-diphenyl -2-picrylhydrazyl assay. Cells grown on BCM resulted in higher (13.73 ± 0.121%) extraction yield than in other media. The phenolic (8.53 ± 0.10 mg/g GAE) and lycopene (0.29 ± 0.008 mg/g) content were highest in cells grown on BCM and BBM, respectively. Microalgae cultured in ANM displayed higher (547.023 ± 34.703 mg/g RE) flavanoid and β-carotene (2.887 ± 0.121 mg/g) content than in other media. Furthermore, cells cultivated in BCM showed highest (97.87 ± 0.088%) scavenging activity than in BBM. These results indicated that the BCM and ANM can be used as alternative to expensive synthetic media for antioxidant production in *C. vulgaris*.

**Keywords:** antioxidants; low cost media; *Chlorella vulgaris*; microalgae

## 1. Introduction

Different metabolic processes in living organisms may produce reactive oxygen species (ROS) that include free radicals such as lipid peroxides, hydrogen peroxide ($H_2O_2$), hydroxyl radicals ($^\bullet OH$) and superoxide ion ($O_{2-}$) [1–3]. Other sources of ROS are external stressors like smoking, chewing tobacco and excessive exposure to sunlight [4]. To protect themselves from the potentially deleterious effects of ROS, animal and plant cells have evolved both enzyme-mediated (e.g., superoxide dismutase, catalase and ascorbate peroxidase) and non-enzymatic antioxidants e.g., ascorbate, glutathione, α-tocopherol, carotenoids and flavonoids [5,6]. However, when ROS production exceeds antioxidant activities damage may occur to molecules such as lipids, proteins, cell membranes, or nucleic acids [1,7,8]. The oxidative damage may trigger various chronic diseases, such as atherosclerosis, cancer, ischemia, diabetes, ageing, and other degenerative diseases in humans, but it should not be considered the primary cause of these diseases [9,10].

Many synthetic antioxidants such as butylated hydroxylanisole (BHA), butylated hydroxyltoluene (BHT), α-tocopherol and propyl gallate have been used in the food industry to reduce food quality deterioration [11]. However, due to their side effects such as liver damage and carcinogenesis as well as a general consumer rejection of synthetic food additives, interest in natural antioxidants is continuously increasing [12,13]. Therefore natural sources of antioxidants are increasingly being sought, particularly from plants, including algae, to identify new and safe candidates.

*Chlorella vulgaris* is a unicellular green microalga measuring about 2 to 10 µm in diameter that can be found in both fresh and marine waters [14]. This alga is the most important species in the microalgal industry; it is cultivated and sold widely as aquaculture feed, health food and nutraceutical [15]. The authors previously reported that many of the chemical components of chlorella such as phenolic compounds, β-carotene, ascorbic acid and tocopherols exhibit antioxidant properties [16]. Ethanol extract of *C. vulgaris*, for instance, exhibited higher antioxidant activity than extracts from other algae such as *Porphyridium cruentum* and *Phaeodactylum tricornutum*, and synthetic antioxidants such as BHA and BHT [17]. Furthermore, growth of colon cancer has been shown to be inhibited by carotenoids extracted from *Chlorella ellipsoidea* and *Chlorella vulgaris* [18]. Other studies have reported that lycopene from the microalgae *Chlorella marina* extracts, has significantly reduced the proliferation of prostate cancer in mice [19]. In addition to anticancer activity, supplementation of *Chlorella* for six weeks has been shown to improve the antioxidant status of male smokers by increasing their plasma vitamin C, α-tocopherol, and erythrocyte catalase and superoxide dismutase (SOD) activities [20].

Although microalgae have received much attention as potential sources of natural antioxidants [21], it is very challenging to obtain affordable antioxidants from microalgae, due to their high cultivation costs [22]. Mass production of microalgae is frequently carried out by using synthetic chemicals as nutrient sources which are very expensive and less accessible in the local market [22]. Studies by [23] have shown costs of culture media are estimated at about 50% of the total biomass production cost and can, therefore, make suitable microalgae commercial production an economically infeasible process. Therefore, for microalgae to become an economically viable source of antioxidants, it is important to reduce the biomass production costs. Many studies that were focused on the antioxidant potential of microalgae were done by using the expensive synthetic media [24–26]. To contribute towards addressing such situation, the purpose of the present study was to assess the influence of low cost media composed of banana stem compost (BCM) and aquaculture wastewater supplemented with 1.0 g/L NPK (ANM) on the content of total phenolics and flavanoids, β-carotene, lycopene and DPPH free radical scavenging activity of *C. vulgaris*.

## 2. Materials and Methods

### 2.1. Preparation of Growth Media and Analysis of Its Nutrients

The ANM medium was prepared by supplementing African catfish pond wastewater with 1.0 g/L NPK (20:20:10) fertilizer. The aquaculture wastewater was collected from an African catfish pond with a stocking density of 6 fish/$m^2$ located at the University of Dar es Salaam Department of Aquatic Science and Fisheries in Kunduchi (Dar es Salaam, Tanzania). Before it's use, the wastewater was filtered using 44 µm filters to remove all macroparticles and sterilized by autoclaving for 15 min at 121 °C. Such prepared media was stored in a refrigerator (4 ± 2 °C) until used for microalgae culture. On the other hand, the BCM medium was prepared by diluting 5% banana (*Musa paradisiaca* Linnaeus) stem compost extract with distilled water. The banana pseudo-stems were collected from a local farm in Dar es Salaam, Tanzania and transported to the laboratory at the University of Dar es Salaam where the outer layers (epidermis) were manually removed by peeling and the remainder was left to decompose for two weeks. Tap water was added to the sliced compost and blended. Such blended material was filtered using 44 µm mesh and the obtained extract sterilized by autoclaving (MLS-3750, SANYO, Japan) at 121 °C for 15 min and stored in a refrigerator (4 ± 2 °C) until used. The ANM and BCM composition of ammonium ($NH_4+$) and phosphorus (P) were analysed using the indophenol blue and ascorbic acid method respectively, following the procedures described by [27]. Nitrate ($NO_3^-$) was analysed using cadmium reduction method as described by zinc (Zn) and copper (Cu) were determined using Atomic Absorption Spectrophotometer (AA240 Varian, Palo Alto, CA, USA) (Table 1).

**Table 1.** Nutrient composition of *C. vulgaris* grown in different media used.

| Parameter | Concentration (mg/L) | | |
|---|---|---|---|
| | BBM | BCM | ANM |
| $NO_3^-$ | 45.591 | 5.707 ± 0.007 | 28.757 ± 0.227 |
| $NH_4^+$ | - | 2.942 ± 0.026 | 45.722 ± 0.028 |
| P | 13.29 | 9.086 ± 0.043 | 11.465 ± 0.012 |
| $K^+$ | 20.987 | 315 ± 0.732 | 350.6 ± 1.0263 |
| $Na^+$ | 20.579 | 27.5 ± 0.577 | 294.467 ± 6.389 |
| $Ca^{2+}$ | 1.703 | 154.133 ± 3.287 | 392.333 ± 0.635 |
| $Zn^{2+}$ | 0.008 | 70.917 ± 2.321 | 51.067 ± 0.133 |
| $Mn^+$ | 0.002 | 15.15 ± 0.465 | 4.487 ± 0.134 |
| $Mg^{2+}$ | 1.84 | 168.617 ± 2.093 | 270.8 ± 1.155 |
| $Cu^{2+}$ | 0.002 | 13.317 ± 0.3 | 4.747 ± 0.237 |
| $Fe^{2+}$ | 0.183 | 42.133 ± 0.765 | 53.333 ± 0.067 |
| $Co^{2+}$ | 0.0004 | - | - |
| $B^{3+}$ | 0.199 | - | - |
| $Mo^{4+}$ | 0.002 | - | - |

## 2.2. Experimental Setup

*Chlorella vulgaris* was obtained from the Department of Botany, University of Dar es Salaam, in Tanzania, where it was isolated from the fish pond using a standard plating method. The species was batch cultured in triplicate for 20 days in 2 L conical flasks using BBM (control), BCM and ANM growth media. The cultures were maintained under laboratory conditions with temperature of 28 ± 1 °C, light intensity 5000 ± 10 lux (measured using a VXLM-636 light meter, Vertex, San Diego, CA, USA), photoperiod (16:8 light: dark cycle) and stirred by continuous aeration. The pH was adjusted to between 9 and10 (AB 15 Accumet Basic, Fisher Scientific, Singapore) by adding 5 M sodium hydroxide or 1 M hydrochloric acid. The microalgae biomass was harvested prior to attaining the stationary growth phase by centrifugation at 978.02× *g* for 10 min. The biomass was thereafter rinsed with distilled water, air-dried and stored at 4 °C until further analysis.

## 2.3. Preparation of the Extracts

The extraction of microalgae samples was done by using 99% ethanol. A precisely weighed 1 g of ground air-dried *C. vulgaris* biomass was mixed with ethanol (100 mL) in a conical flask. The mixture was placed in a dark place at room temperature (24 °C) for 24 h and stirred during the extraction time to ensure complete extraction [28]. After extraction, the samples were filtered with Whatman No. 4 filter paper in a Büchner funnel under vacuum to remove the algal material. The solvents were then evaporated at 50 mm Hg pressure and 50°C using a rotary evaporator (Heidolph Instruments, Gmbh & Co. KG, Schwabach, Germany) and the dark green viscous mass was obtained. After that, the obtained extracts were weighed and refrigerated at −20 °C until analysis. Extraction yield, total phenolic content (TPC), total flavonoid content (TFC) and antioxidant activity (DPPH assay) were evaluated for the obtained extracts, as described below. All experiments were performed in triplicate.

## 2.4. Determination of the Extraction Yield

The extraction yield of *C. vulgaris* at different cultivation media was calculated using Equation (1) shown below:

$$\text{Yield } (\%) = \frac{W_1 \times 100}{W_2} \tag{1}$$

where $W_1$ is the weight of extract after evaporation of ethanol and $W_2$ is the dry weight of algae.

## 2.5. Determination of Total Phenolic Content

The total content of phenolic compounds of the ethanolic extracts was determined spectrophotometry using the Folin-Ciocalteu reagent as described by Kähkönen et al. [29], and gallic acid was used as a standard phenolic compound. Each evaporated thick and viscous ethanolic extract was diluted with 20 mL methanol. Briefly, 0.5 mL of extract was introduced into a test tube and then mixed thoroughly with 2.5 mL of Folin-Ciocalteu reagent. The mixture was allowed to react for 3 min after which 2 mL of 7.5% (*w/v*) sodium carbonate was added. The mixtures were agitated with a vortex mixer and incubated for 30 min at room temperature (25 °C) in the dark. The absorbances of the sample and a prepared blank were measured using a spectrophotometer (UV 6305, Genway, London, UK) at a wavelength of 765 nm. Quantification was done based on the standard curve of gallic acid. The total phenolic content was expressed as milligram of gallic acid equivalents (GAE) per gram dry weight of microalgae (mg GAE/g DW).

## 2.6. Determination of Total Flavonoid Content

The total flavonoid content was determined by calorimetric assay according to the method described by Serra Bonveni et al. [30]. In short, 3 mL of extract was pipetted into a test tube and mixed with 0.3 mL of 2% (*w/w*) $AlCl_3$ in a methanol solution containing 5% acetic acid, using a vortex mixer. The absorbance was read immediately at 425 nm using a spectrophotometer (UV 6305, Genway). The absorbance of a prepared blank was also recorded. Rutin was used as a standard with methanol as blank. The total flavonoids content was expressed as rutin equivalents in milligrams per grams of the dry weight of microalgae (mg RE/g DW).

## 2.7. Determination of β-Carotene and Lycopene

β-Carotene and lycopene were determined according to the method described by Nagata et al. [31]. Briefly, air dried *C. vulgaris* biomass was crushed using a mortar and pestle and then weighed. 100 mg was mixed with 10 mL of acetone-hexane at a ratio of 4:6, respectively. After that, the mixture was shaken vigorously for 1 min and filtered through Whatman No.4 filter paper. The absorbance of the filtrate was measured at 453, 505 and 663 nm using a spectrophotometer (UV 6305, Genway). Contents of β-carotene and lycopene were calculated using Equations (2) and (3), and results were presented as mg/g of extract:

$$\text{Lycopene (mg/ 100 mL)} = -0.0458A_{663} + 0.372A_{505} - 0.0806A_{453} \tag{2}$$

$$\text{β-carotene (mg/100 mL)} = 0.216A_{663} - 0.304\,A_{505} + 0.452A_{453} \tag{3}$$

## 2.8. Determination of Antiradical Activity against DPPH Radical

The ability to capture free radicals from microalgae extracts was evaluated using the (1,1-diphenyl-2-picrylhydrazyl (DPPH) assay [32]. This method is simple, rapid and convenient, independent of sample polarity for the screening of many samples for radical scavenging activity [33]. Briefly, a series of extract concentrations with different ratios of extracts to methanol, i.e., 1: 0.1, 1:0.01, 1:0.001, 1:0.0001 and 1:0.00001 were prepared. For each 4.9 mL of diluted extract, 0.1 mL of 5 mM DPPH radical in methanol solution was added. Then the mixtures were shaken to ensure that all the solutions were well mixed and placed in dark condition at room temperature for 30 min. After that, the absorbance of each sample of algae extract containing DPPH ($A_1$), without DPPH ($A_s$) and only DPPH solution without algae extract ($A_o$, called control) were recorded at 517 nm using a spectrophotometer (UV 6305, Genway). Percentage inhibition by DPPH in samples was determined by comparison with the methanol-treated control group and calculated using the following formula:

$$\text{DPPH radical scavenging activity (\%)} = \left[\frac{A_0 - (A_1 - A_s)}{A_0}\right] \times 100 \tag{4}$$

where $A_o$ is the absorbance of the control solution (containing only DPPH), $A_1$ is the absorbance of test sample containing DPPH solution and $A_s$ is the absorbance of the test sample without DPPH. Ascorbic acid (Vit C) was used as a standard. The concentration of *C. vulgaris* extract (mg/mL) required for decreasing the initial concentration of DPPH by 50% ($EC_{50}$) was also calculated and expressed relative to ascorbic acid in terms of ascorbic acid equivalent.

### 2.9. Statistical Analyses

All the results were calculated as a mean value ± the standard error (SE). The computer software R version 3.6.3 was used to analyze the data. Normality of data was tested using Shapiro-Wilk's (SW) tests, and homogeneity of variance using Levene's tests (LT). When the SW and LT showed $p > 0.05$, one way ANOVA was used to test the data. If SW and LT showed $p < 0.05$, data were either square root or log transformed. Where normality and homogeneity tests still did not pass, Kruskal–Wallis tests were used instead of ANOVA test. Differences were considered significant when $p$ values were below 0.05.

## 3. Results

### 3.1. Extraction Yield

Extraction yields of the microalgae at different media are presented in Figure 1a. There was a significant dependence of *C. vulgaris* extraction yield on the cultivation media (F = 1577.9, df = 2, $p = 0.000$). Microalgae cultured in BCM had extraction yield (13.73 ± 0.121%) that was significantly higher ($p < 0.0001$) than in BBM (10.90 ± 0.052%) and ANM (6.87 ± 0.072%) respectively. Also, microalgae grown in BBM showed higher extraction yield compared to those raised in ANM ($p < 0.000$).

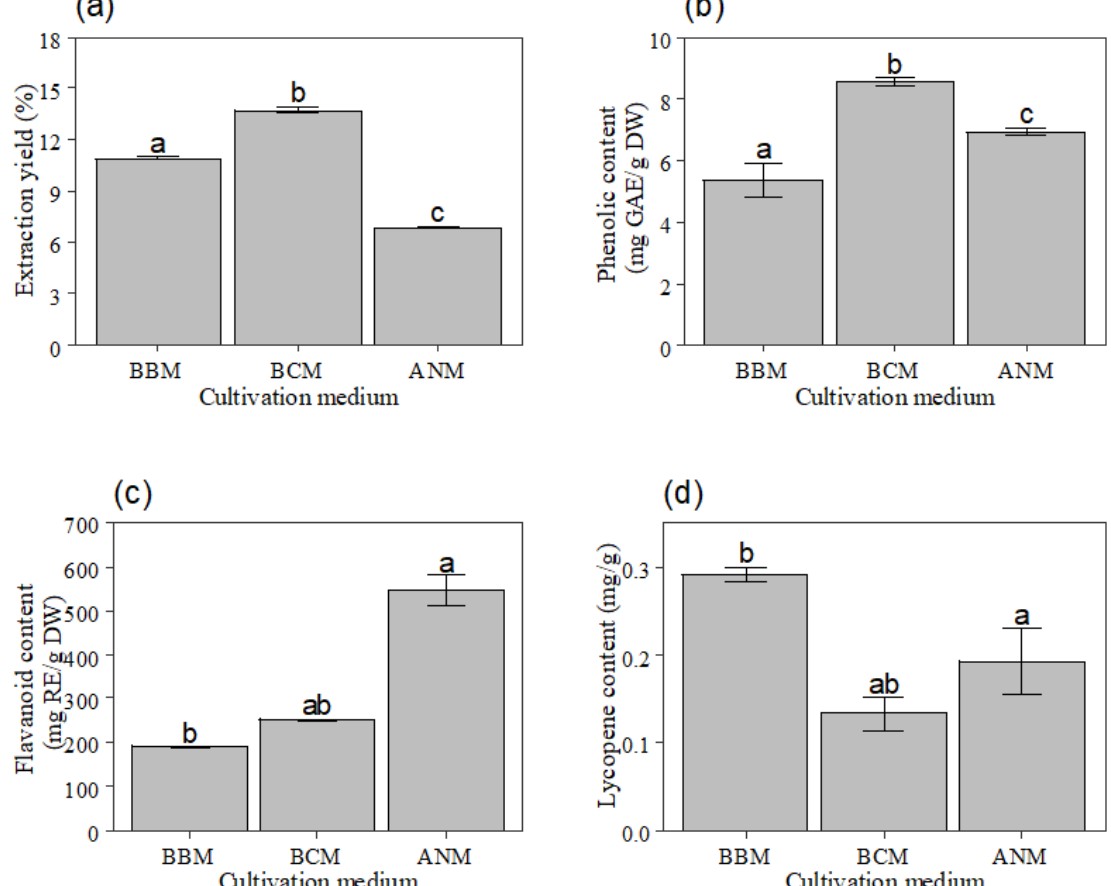

**Figure 1.** *Cont.*

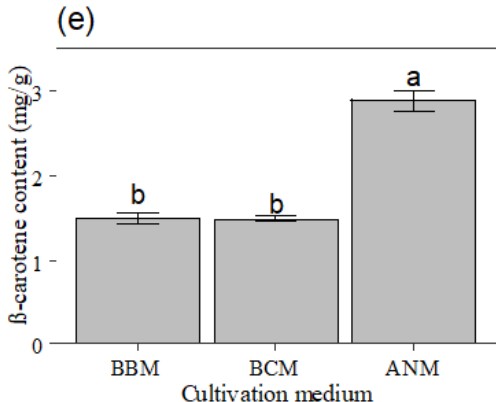
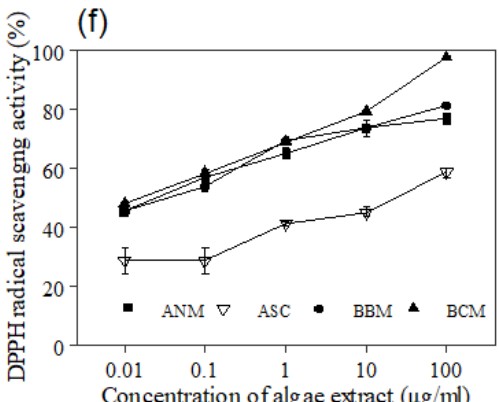

**Figure 1.** (**a**) Extraction yield; (**b**) Phenolic content; (**c**) Flavanoid content; (**d**) lycopene content; (**e**) β-carotene content and (**f**) DPPH radical scavenging activity of *C. vulgaris* grown in different media. (Means sharing the same letter are not significantly different).

### 3.2. Total Phenolic Content

A significant variation in the total phenolics content of *C. vulgaris* was noticed between growth media (F = 22.602, df = 2, $p$ = 0.002; Figure 1b). In general, cells grown in BCM contained higher amounts of phenolic content (8.53 ± 0.10 mg/g GAE) when compared to BBM (5.352 ± 0.556 mg/g GAE) and ANM (6.92 ± 0.13 mg/g GAE), respectively ($p$ < 0.05). Also, a significantly higher ($p$ = 0.037) amount of phenolic compounds was observed in *C. vulgaris* raised in ANM than in BBM.

### 3.3. Total Flavonoid Content

The results revealed that there was a significant effect of the growth media on the *C. vulgaris* flavonoid content of ($x^2$ = 7.2, df = 2, $p$ = 0.02732; Figure 1c). Microalgae cultured on ANM displayed significantly higher ($p$ = 0.022) flavonoid content (547.023 ± 34.703 mg/g RE) than observed in BBM (191.296 ± 2.904 mg/g RE). On the other hand, no significant variation ($p$ > 0.05) was recorded on the amount of flavanoid content in cells grown in BBM and BCM (251.9938 ± 3.293 mg/g RE) as well as between BCM and ANM.

### 3.4. β-Carotene and Lycopene

Both total β-carotene (F = 92.728, df = 3, $p$ = 0.000; Figure 1e) and lycopene (F = 10.526, df = 2, $p$ = 0.011; Figure 1d) content of *C. vulgaris* showed significant variation between media. The β-carotene content of *C. vulgaris* grown in ANM (2.887 ± 0.121 mg/g) was significantly higher ($p$ < 0.05) than 1.494 ± 0.073 mg/g and 1.479 ± 0.034 mg/g recorded in BBM and BCM respectively. On the other hand, no significant variation in β-carotene content of *C. vulgaris* was observed in BBM and BCM ($p$ = 0.9912). Furthermore, *C. vulgaris* grown in BBM (0.29 ± 0.008 mg/g) had statistically higher lycopene content than BCM (0.133 ± 0.019 mg/g). Furthermore, no significant variation was recorded in lycopene content of cells grown on BBM and ANM also between BCM and ANM ($p$ > 0.05).

### 3.5. Free Radical Scavenging Using DPPH Assay

DPPH radical scavenging activity showed microalgae concentration dependency (F = 127.5, df = 4, $p$ = 0.000; Figure 1f) and the highest value were recorded in 100 µg/mg of ethanolic extract. Moreover, microalgae cultivated in BCM displayed significantly higher scavenging activity (97.9 ± 0.1%) than in BBM (81.29% ± 0.088), ANM (77.07 ± 1.657%) and standard ascorbic acid (58.648 ± 1.596%), respectively ($p$ = 0.000). However, no significant variation ($p$ = 0.686) in antioxidant activity was recorded in *C. vulgaris* raised in BBM and ANM. Also, there was a significant variation of EC50 values in Chlorella ethanolic extracts and standard ascorbic acid ($x^2$ = 8.1439, df = 3, $p$ = 0.04313). The lower EC50 value means the more powerful antioxidant capacity. The standard ascorbic acid had statistically

higher (11.066 ± 0.365 μg/mg) EC50 value than observed in BBM (0.011 ± 0.000 μg/mg), ANM (0.011 ± 0.000 μg/mg) and BCM (0.01 ± 0.000 μg/mg), respectively. Also, no significant variation was observed in $EC_{50}$ values for cells grown in BBM relative to those grown in BCM and ANM ($p > 0.05$).

## 4. Discussion

### 4.1. Chlorella Vulgaris Extraction Yield

Findings by previous researchers indicated that extraction yield depends on solvent the solvent used, pH, extraction time and temperature as well as on the chemical compositions of the sample [34–36]. A study by [22] has reported significant differences in extraction yield in cyanophytic algae reared using different growth media. The difference in *C. vulgaris* extraction yield in this study implies that the different cultivation media used could have affected the extraction yield. The lower extraction yield recorded in ANM may be due to its high carotenoids contents, as it was reported that, carotenoids are lipophilic compounds [24], making their extraction difficult with polar solvents such as ethanol [37]. As compared to results of the present study, [38] observed almost similar yield (11.8 ± 0.000%) in mangosteen (*Garcinia mangostana* Linn), although a recent study by [22] detected higher yield (21.63 ± 0.04%) in extracts from Spirulina (*Arthrospira fusiformis*) using ethanol as a solvent. These differences in extraction yield could be explained by variations in the chemical composition of the sample.

### 4.2. Total Phenolic Contents

Phenolic compounds are considered one of the most effective natural antioxidants found in microalgae particularly *Chlorella* sp., *Nannochloropsis* sp. and the cyanophtic alga *Spirulina platensis* [39,40]. These compounds serve as important antioxidants due to their ability to donate a hydrogen atom or an electron to form stable radical intermediates. According to [41], phenolic compounds are integral structural components of microalgae cell walls which are involved in chemical protective mechanism against stresses of biotic factors (e.g., grazing, settlement of bacteria or other fouling organisms) and abiotic (e.g., ultraviolet radiation and metal contamination). In addition, due to their antioxidant activity, phenolic compounds possess diverse biological activities, such as anticancer, antioxidative, antibacterial, anti-allergic, anti-diabetes, anti-ageing, anti-inflammatory and anti-HIV activities [42,43]. Total phenolic contents in microalgae seem to depend on many factors including algal species, the area of cultivation, seasonal, physiological, and environmental variations [44]. The phenolic content in the analyzed *C. vulgaris* was approximately the same as reported for *Chlorella prototliecoides* (14.35 mg GAE/g) and *Chlorella pyrenoidosa* (13.90 mg GAE/g) in hexane fractions [45]. On the other hand, Reference [16] reported much lower phenolic content (1.44 ± 0.04 mg tannic acid equivalent/g of algae powder) in *Chlorella* aqueous extract. Our results are also lower than the published data (65.89–258.83 mg GAE/g) in the *Chlorella* methanolic extracts [46]. The differences between phenolic contents could be due to differences in the solubility of phenolic compounds in different extraction solvent systems and constituent variance among algae species [1,16,24].

### 4.3. Flavanoid Content

Carotenoids together with flavonoids appear to prevent UV-induced sunburn in humans with light-sensitive skin containing low levels of melanin [47]. As can be seen in Figure 1c, the present study found *C. vulgaris* to be a rich source of flavanoids and the highest content was observed in ANM, followed by BCM and BBM. The current results are higher than the finding of [22] who obtained the highest flavanoid content of 13.25 ± 0.5 RE mg/g for the cyanophyta *Arthrospira fusiformis* ethanolic extract in their study on antioxidants activity of the cyanobacterium, *Arthrospira* (Spirulina) *fusiformis* cultivated in a lowcost medium. Furthermore, [38] reported flavanoid content of Thai indigenous plants in the range from 1.6 ± 0.0 to 50.4 ± 0.3 mg RE/g DW which are lower than the present results. This confirms the suitability of microalgae as a good source of natural antioxidants.

### 4.4. β-Carotene and Lycopene

It is well-known that microalgae are one of the most important natural sources of carotenoids, which serve as sources of antioxidants and function as pro-vitamin A [48]. The beneficial effects of natural carotenoids (lutein, β-carotene, lycopene) to human and animal health have been reported from the numerous clinical and epidemiological studies in various populations [49,50]. β-carotene as a major source of vitamin A is necessary for the functions of the retina and affects many tissue types [49,50]. Besides this, it was also reported to contain anticancer, antiaging, anti-inflammatory, antibacterial and antiviral actions. Reference [51] have shown that β-carotene and lycopene inhibited the growth of human breast cancer cells due to their antioxidant activity. Furthermore, References [52] and [53] found that various in vivo and in vitro studies on the effect of lycopene in tumor cell lines have displayed the inverse relationship between lycopene intake and prostate cancer/other tumor cell growth. The finding of this study supports the view that the nutrient composition of the growth medium has an important influence on carotenoids content and composition in microalgae biomass [54]. The ß-carotene and lycopene were highest in ANM and BBM respectively, compared to other growth media. The results obtained in our study fall within the range reported by [54] who obtained carotenoids content of $0.25 \pm 0.09$ to $3.04 \pm 0.20$ (mg·g$^{-1}$ DW) for *Chlorella* sp.

### 4.5. Free Radical Scavenging Using DPPH Assay

Antioxidants play an important role in protecting cell tissues from free radicals, thus protecting the living organism against infections and degenerative diseases such as cancer. The antioxidant activity of the extracts indicates the presence of compounds that can interact with free radicals and act as electron donors [46]. DPPH is a well-known compound that has been used extensively as a free radical to evaluate the radical scavenging abilities of natural compounds [55]. DPPH is a stable free radical that has a strong absorption band centered at about 517 nm, which is diminished in the presence of antioxidants capable of reducing it to its hydrazine form by hydrogen/electron donation [56]. In the present finding, all tested microalgae extract scavenged DPPH at different levels. Observing the phenolic content data, a positive correlation between total phenolic content and antioxidant activity was noted. Therefore, high scavenging activity in BCM extract could be probably due to its higher content of phenolic compounds compared to BBM and ANM extracts. On the other hand, [57] in their study on antioxidant activity of three edible seaweeds from two areas in South East Asia, reported a similar correlation after using the DPPH radical scavenging assay. These authors found a decrease in IC$_{50}$ with an increase in the total phenolic content level. Investigations revealed that the high amount of polyphenolic compounds present in the seaweed were capable of functioning as free radical scavengers. Our findings are in agreement with [21], who presented results of DPPH radical-scavenging activity of $96.8 \pm 0.07$ % in a red alga, *Polysiphonia urceolata* crude extract, using an ascorbic acid solvent fraction. On the other hand, the current results differ from the findings of [58] who found the highest radical scavenging ability of $70.3 \pm 0.6$ % in *Nannochloropsis oculata* hexane extracts. This difference might be due to the differences in the chemical nature of the compounds that contribute to antioxidant responses within the cellular structure of these species [24].

### 5. Conclusions

Overall, this study shows that the cultivation media significantly affected the antioxidant compounds production and the DPPH scavenging ability of *C. vulgaris* extracts. The highest antioxidant contents and free radical scavenging ability was observed in microalgae grown on BCM and ANM media compared to BBM. The present findings suggest that BCM and ANM are very potential in minimizing the production costs of natural antioxidants from *C. vulgaris* and could substitute the expensive synthetic media. The new information presented herein are useful to support more consumption of *C. vulgaris* products as a source of natural antioxidants and decrease the risk of degenerative disease

conditions such as cancer, diabetes, ageing, inflammation, and stroke and neurodegenerative disease in living organisms.

**Author Contributions:** Conceptualization, K.M., M.S.K. and M.S.P.M. methodology, K.M., M.S.K. and M.S.P.M.; software, K.M.; validation, K.M., M.S.K. and M.S.P.M.; formal analysis, K.M.; investigation, M.S.K. and M.S.P.M.; resources, M.S.K. and M.S.P.M.; data curation, K.M., M.S.K. and M.S.P.M.; writing—original draft preparation, K.M.; writing—review and editing, K.M., M.S.K. and M.S.P.M.; visualization, K.M., M.S.K. and M.S.P.M.; supervision, M.S.K. and M.S.P.M.; project administration, M.S.K. and M.S.P.M.; funding acquisition, M.S.K. and M.S.P.M. All authors have read and agreed to the published version of the manuscript.

**Funding:** The study was funded by the Swedish International Cooperation Agency (Sida) through the Bilateral Marine Science Program of 2015–2020 (grant No. 51170071).

**Acknowledgments:** We wish to thank the Institute of Marine Science, University of Dar es Salaam for financial support through the Sida (Swedish International Development Cooperation Agency) supported Bilateral Marine Sciences Program.

**Conflicts of Interest:** The authors declare to have no conflict of interest.

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
