# Peer review of "Assessment of Antioxidant Contents and Free Radical-Scavenging Capacity of Chlorella vulgaris Cultivated in Low Cost Media"

_applsci, doi:10.3390/app10238611_

Round 1

Reviewer 1 Report

This interesting experimental study aimed to find out antioxidants contents in  Chlorella vulgaris that was grown in low-cost media.

There are several comments, which can be taken in the consideration to improve the text.

  • Lines 10-12. Could be placed into Introduction.
  • Lines 11-12. Please, clarify, which tissues are you writing about?
  • Line 11 : “several synthetic antioxidants” . What do you mean? Provide some examples, please.
  • Line 12:” associated adverse side”. Please, clarify.
  • Lines 18-22. Please, indicate how much higher measured parameters (cell growth, the phenolic and lycopene content, scavenging activity) were detected. It is not necessary to show “p < 05” in the Abstract. You can point it in the Result section.
  • Please, check the text for misprints. For example, line 233: instead of “cyanophtic alga” has to be “Cyanophyta algae”.
  • How authors can explain algae growth without Co2+, B3+ and Mo4+ in BCM and ANM growth mediums?
  • Please, clarify what you mean as an “extraction yield”.
  • Line 228. Spirulina has to be written starting with a capital S.
  • What hypothesis and author’s ideas could be about the possible reasons for the statement about “The highest antioxidant contents and free radical scavenging ability” algae culture that was grown in BCM and ANM?

Author Response

Response to reviewers

Journal: Applied Sciences (ISSN 2076-3417)

Title: Assessment of Antioxidant Contents and Free Radical-Scavenging Capacity of Chlorella vulgaris Cultivated in Lowcost Media

Manuscript ID: applsci-992686

Authors:  Kulwa Mtaki *, Margareth S. Kyewalyanga , Matern S.P. Mtolera

Reviewer 1:

Comment 1:

Lines 10-12. Could be placed into Introduction

Response:

We accepted the comment   and deleted lines 10-13 in the abstract. These sentences are present in the introduction section. Please, see page 1 lines 34-43 in the revised manuscript.

Comment 2:

Lines 11-12. Please, clarify, which tissues are you writing about?

Response:

The tissue we meant is living tissue in living organism.

Comment 3:

Line 11: “several synthetic antioxidants” . What do you mean? Provide some examples, please.

Response:

Synthetic antioxidants are created from chemical processes since they do not occur in nature and often added to food as preservative to prevent lipid peroxidation e.g Butylated hydroxytoluene (BHT), butylated hydroxyanisole (BHA), α-tocopherol and propyl gallate. There are also several synthetic antioxidants which are used in the treatment of different diseases e.g. 5-amino-salicylic acid, mexidol, carvedilol and melatonin

Comment 4:

Line 12:” associated adverse side”. Please, clarify.

Response:

We meant the side effects like liver damage and carcinogenesis caused by using the synthetic antioxidant.

Comment 5:
Lines 18-22. Please, indicate how much higher measured parameters (cell growth, the phenolic and lycopene content, scavenging activity) were detected. It is not necessary to show “p < 05” in the Abstract. You can point it in the Result section.

Response:

We accepted the comment and revised it, Please, see page 1, lines 19-23 in the revised manuscript.

Comment 6:

Please, check the text for misprints. For example, line 233: instead of “cyanophtic alga” has to be “Cyanophyta algae”

Response:

We accepted the comment and revised it, Please, see page 8, line 234 in the revised manuscript.

Comment 7:

How authors can explain algae growth without Co2+, B3+ and Mo4+ in BCM and ANM growth mediums?

Response:

The Co2+, B3+ and Mo4+ in BCM and ANM growth media were not analysed, we analysed only few nutrients. But we believe that those elements were present in the media.

Comment 8:

Please, clarify what you mean as an “extraction yield”.

Response:

Extraction yield are solubles dissolved during microalgae extraction using ethanol as a solvent.

Comment 9:

Line 228. Spirulina has to be written starting with a capital S.

Response:

We accepted the comment and made revisions, Please, see page 8, line 229 in the revised manuscript.

Comment 10:

What hypothesis and author’s ideas could be about the possible reasons for the statement about “The highest antioxidant contents and free radical scavenging ability” algae culture that was grown in BCM and ANM?

Response:

The possible reason for the highest antioxidant contents and free radical scavenging ability in microalgae cultivated in BCM and ANM could be due to the high nutrient contents that were present in the media.

Reviewer 2 Report

It is opinion of the reviewer that this paper before acceptance needs several corrections. My individual comments are listed below.

14 - It should be „total phenolics and total flavonoids”.

18 - It should be “1,1-diphenyl …”.

19 – Extraction of what?

29 - It should be “OH”.

34 - It should be “α-tocopherol”.

39 - It should be “… hydroxylanisole … hydroxyltoluene …”.

41 – BHA and BHT are not used for living cells. They are used as synthetic food antioxidants.

49 - It should be “chlorella”.

49 - It should be “β-carotene”>

L 49 - It should be “tocopherols”.

58 – Full name of SOD.

70 - It should be “ content of total phenolics and flavonoids”.

89 - It should be “NO3-“.

89 - It should be “zinc”.

98 – Remove “(NaOH) … (HCl)”.

100 – Centrifugation must be characterized by “x g” instead of “rpm”.

111 - It should be “total flavonoid content (TFC) …”.

107 - It should be “Büchnera”.

107 - Whatman No.4 filter paper

111 - It should be “(DPPH assay)”.

119 - It should be “… as described by Kähkönen et al. [29].

130 - It should be “colorimetric method according to Serra Bonveni et al. [29].

138 - It should be “ described by Nagata et al. [30]”.

145 – It should be “Antiradical activity against DPPH radical”.

151 and other lines - It should be “DPPH radical”.

170 – Sofware used for statistical analysis?

182 – TPC should be reported with two digitals after decimal point.

174 and other lines “”C. vulgaris” must be written with italic.

The results of flavonoids are very controversial. The content of flavonoids should be similar to that of total phenolics – F-C reagent gives reaction with flavonoids!

TFC and TFC in Methods are reported per 100 g. However, in Results per g!

200 , 277 – It should be “…. DPPH assay”.

203 – Results with one digital after decimal point (mean value and SD”.

References – Latin names must be written with italic.

Figure 1 – Significance of differences between mean values must be marked with letters (a for the highest result”.
